# Biopotentials of Collagen Scaffold Impregnated with Plant-Cell-Derived Epidermal Growth Factor in Defective Bone Healing

**DOI:** 10.3390/ma16093335

**Published:** 2023-04-24

**Authors:** Sher Bahadur Poudel, Govinda Bhattarai, Tae-Ho Kwon, Jeong-Chae Lee

**Affiliations:** 1Department of Molecular Pathobiology, New York University College of Dentistry, New York, NY 10010, USA; sbp4@nyu.edu; 2Cluster for Craniofacial Development & Regeneration Research, Institute of Oral Biosciences, Jeonbuk National University, Jeonju 54896, Republic of Korea; govinda@jbnu.ac.kr; 3Natural Bio-Materials Inc., Iksan 54631, Republic of Korea; thkwon@jbnu.ac.kr; 4Research Center of Bioactive Materials, Jeonbuk National University, Jeonju 54896, Republic of Korea

**Keywords:** plant cell suspension culture system, recombinant human EGF, collagen scaffold, defective bone healing, human periodontal ligament cells

## Abstract

The combination of scaffolds with recombinant human epidermal growth factor (rhEGF) protein can enhance defective bone healing via synergistic activation to stimulate cellular growth, differentiation, and survival. We examined the biopotentials of an rhEGF-loaded absorbable collagen scaffold (ACS) using a mouse model of calvarial defects, in which the rhEGF was produced from a plant cell suspension culture system because of several systemic advantages. Here, we showed a successful and large-scale production of plant-cell-derived rhEGF protein (p-rhEGF) by introducing an expression vector that cloned with its cDNA under the control of rice α-amylase 3D promoter into rice calli (*Oryza sativa* L. cv. Dongjin). Implantation with p-rhEGF (5 μg)-loaded ACSs into critical-sized calvarial defects enhanced new bone formation and the expression of osteoblast-specific markers in the defected regions greater than implantation with ACSs alone did. The potency of p-rhEGF-induced bone healing was comparable with that of *Escherichia coli*-derived rhEGF protein. The exogenous addition of p-rhEGF increased the proliferation of human periodontal ligament cells and augmented the induction of interleukin 8, bone morphogenetic protein 2, and vascular endothelial growth factor in the cells. Collectively, this study demonstrates the successful and convenient production of p-rhEGF, as well as its potency to enhance ACS-mediated bone regeneration by activating cellular responses that are required for wound healing.

## 1. Introduction

Implanting autografts or allografts is often required for the recovery of large bone defects [1]. The application of bioactive materials in combination with scaffolds is an alternative strategy for bone regeneration in guided tissue engineering. Many investigators have tried to develop bioscaffolds that can enhance bone regeneration via conjugation with growth factors. Accumulated evidence highlights that the combined application of scaffolds with growth factors provides therapeutic enhancement of bone tissue engineering [2,3,4]. The synergistic effect of growth factors is related to their abilities to regulate various biological processes, including the differentiation, proliferation, migration, and survival of cells [3]. Bone morphogenetic proteins (BMPs), epidermal growth factor (EGF), fibroblast growth factors (FGFs), insulin-like growth factors (IGFs), and platelet-derived growth factor are the factors that enhance the efficacy of bioscaffolds on defective bone healing. Of these factors, EGF is a small single-chain polypeptide that is derived via the proteolytic processing of transmembrane precursor glycoprotein [5]. EGF regulates multiple cellular functions, including cell proliferation, apoptosis, migration, and differentiation through interaction with its specific receptor, EGFR [6]. EGF signaling also plays crucial roles during the development and maintenance of bone and organs [6]. This information suggests that EGF functions as an enhancing factor in the scaffold-mediated healing of defective bones.

As the medicinal and cosmetic use of growth factors has gradually increased, it is of great importance to enhance the production efficacy and pharmaceutical activity of growth factors without side effects. Regarding this, the use of a transgenic plant cell suspension culture system is to be an attractive approach in producing recombinant proteins [7,8]. Compared with the DNA recombinant technology systems utilizing *Escherichia coli* (*E. coli*) and other heterologous expression systems (mammalian cell, insect, and yeast cultures), plant cell culture systems have several advantages, such as inexpensive production costs and large-scale production [9]. The cultured plant-cell-derived proteins could also be conveniently purified without contamination risks from viral and bacterial toxins [9,10]. Similarly, we have demonstrated that the rice calli suspension culture is a powerful system used to produce biologically active recombinant human (rh) IGF1 and rhFGF2 [11,12]. The implantation of absorbable collagen scaffolds (ACSs) loaded with the plant-derived rhIGF1 or rhFGF2 enhanced defective bone healing more greatly than the implantation of ACSs alone did [11,12]. Furthermore, the synergistic activity of plant-derived factors on bone regeneration was comparable with that of *E. coli*-derived factors. 

Here, we explored whether the transgenic plant cell suspension culture system is also useful to produce a biologically active rhEGF using the rice amylase 3D (RAmy3D) expression system as described previously [11,12,13,14]. This is because the utilization of the RAmy3D system in a rice cell suspension medium produced greater levels of growth factors compared with those expressed in transgenic tobacco plastid, tobacco cell culture, and soy and maize seed systems [13,14]. Our current findings supported the concept that the rice calli suspension culture might be a promising system to produce plant-derived rhEGF (p-rhEGF). The implantation of p-rhEGF-impregnated ACSs facilitated greater bone regeneration in a mouse model of calvarial defects than the implantation of ACSs alone did. The direct addition of p-rhEGF also stimulated cellular responses that are required for wound healing in human periodontal ligament (hPDL) cells. Moreover, we found that in vivo biopotentials of p-rhEGF were comparable with those of *E. coli*-derived rhEGF protein (e-rhEGF). Collectively, the current findings along with the previous studies [11,12] confirm the clinical usefulness of ACSs on defective bone healing and their enhancement via combination with growth factors. 

## 2. Materials and Methods

### 2.1. Chemicals and Laboratory Consumables

The e-rhEGF was purchased from BioVision (Cat. No. 4022-1000; Milpitas, CA, USA). Antibodies specific to runt-related transcription factor-2 (Runx2; ab23981) and osteopontin (OPN; ab8448) were obtained from Abcam (Cambridge, UK), while anti-osteocalcin (OCN; BS7961) was from Bioworld Technology Inc. (St. Louis Park, MN, USA). Fetal bovine serum (FBS) was purchased from HyClone Laboratories (Logan, UT, USA). Unless specified otherwise, other chemicals and laboratory consumables were purchased from Sigma-Aldrich Co. LLC (St. Louis, MO, USA) and Falcon Labware (Becton-Dickinson, Franklin Lakes, NJ, USA).

### 2.2. Production of p-rhEGF Using a Plant Expression System

We produced p-rhEGF using the RAmy3D promoter in transgenic rice cell suspension culture system as described previously [12,15,16]. In brief, a sequence including the human EGF gene harboring the signal peptide of RAmy3D was synthesized using the human EGF DNA sequence (GenBank accession No. NM 001178130). After the processes for codon optimization, cloning, and amplification, the resultant PCR product containing the mature peptide region of the human EGF gene and the signal sequence of RAmy3D were introduced into a plant expression vector, pMYN75, fused with hygromycin phosphotransferase (HPT), which is a selection marker for plant transformation (Figure 1). 

We transformed the human EGF-pMYN75 with rice calli (*Oryza sativa* L. cv. Dongin) using the particle-bombardment-mediated transformation technique [16]. The explants were then transferred to N6 selection medium supplemented with casein enzymatic hydrolysate (0.3 g/L), 2,4-dichlorophenoxyacetic acid (2 mg/L), gelite (2 g/L), glutamine (0.5 g/L), hygromycin B (35 mg/L), proline (0.5 g/L), and sucrose (30 g/L) by replacing them with the same fresh media at 2–3-week intervals. HPT-resistant calli and more than 16 transgenic calli were incubated in sucrose-free N6 medium for three days followed by the quantification of p-rhEGF protein using a human EGF-specific enzyme-linked immunosorbent assay (ELISA) kit (Endogene, Woburn, MA, USA). Finally, we selected the S131-2 callus as the optimum cell line to efficiently produce p-rhEGF in N6S medium. To establish the cell suspension culture, the S131-2 callus was grown in 1 L of gelite-free N6S medium at 25 °C in a shaking incubator, and the inocula (200 mL) were transferred to new media every nine days. To induce the expression of the human EGF gene, media were removed from the suspension culture via aspiration, and sucrose-free N6S medium was added to the cultures. To evaluate a time-dependent production phase of p-rhEGF, the supernatant of the S131-2 callus was harvested via filtration at various times of incubation, and the protein level was quantified using a Bradford protein assay (Bio-Rad, Hercules, CA, USA). The maximum production level (approximate 33 mg/L) of p-rhEGF in the callus was reached 13 days post-incubation (Figure 2).

The products including p-rhEGF were purified via hydrophobic interaction chromatography and anionic exchange chromatography, followed by identification via sodium dodecyl sulfate–polyacrylamide gel electrophoresis (SDS-PAGE) and high-performance liquid chromatography (HPLC). The final product of p-rhEGF showed purity of more than 95%. The p-rhEGF was lyophilized and stored at −70 °C before use.

### 2.3. Fabrication of ACSs

We used an ACS as a carrier of rhEGF because of its bone-like porosity and general use in animal models of critical-sized bone defects. ACSs were fabricated using type I atelocollagen powder (KOKEN Corp., Osaka, Japan) according to the methods described previously [17]. Briefly, the 50 mM acetic acid solution containing 10 mg of collagen powder/mL was coprecipitated with chondroitin-6-sulfate (5 mg/mL) in a stirring homogenizer. The collagen-chonbroitin-6-sulfate mixture was lyophilized at −80 °C for 6 h to yield ACSs. Lyophilized ACSs were incubated in 20 mL of 40% (*v/v*) ethanol containing 50 mM 2-morpholineoethane sulfonic acid (MES; Fluka Chemie, Buchs, Switzerland) (pH 5.5) for 30 min at −25 °C followed by the reaction process in 40% (*v/v*) ethanol containing 24 mM 1-ethyl-3-(3-dimethyl aminopropyl), 5 mM N-hydroxysuccinimide, and 50 mM MES for 12 h. Thereafter, ACSs were washed step-by-step with 0.1 M Na_2_HPO_4_ (pH 9.0) for 12 h, 1 M NaCl for 12 h, and 2 M NaCl for 48 h. Finally, the ACSs were rinsed with distilled water, lyophilized, and sterilized with 10 kGy of γ-irradiation.

### 2.4. Charactization of ACSs

The mechanical properties of ACSs were evaluated using a mechanical tester (BS5669, INSTRON, Norwood, MA, USA) by subjecting them to uniaxial compressive loading with a crosshead speed of 0.5 mm/min. While the value of compressive strength was determined based on the strain stress (10 to 20% strain), the modulus of elasticity was calculated using linear regression from the region of the stress–strain curves. The degradation property of ACSs was also determined by incubating circularly cut ACSs in 1 mL of phosphate-buffered saline (PBS) for 4 weeks. During the incubation, PBS was replaced with a new solution every 3 days. The degradation rate of ACSs was calculated by applying the following equation: Weight loss (%) = (W_0_ − W_1_/W_0_) × 100. In addition, field emission scanning electron microscopy (FE-SEM; Gemini500, Carl Zeiss, Oberkochen, Germany) was applied to obtain the surface images of ACSs after sputter-coating them with platinum under a vacuum. 

### 2.5. Animals and Ethical Statements 

Male C57BL/6 mice (6 weeks old) were obtained from Orient Bio (Daejeon, South Korea) and assigned randomly to experimental groups. Among the groups, the mean body weights did not differ, and all the mice were equilibrated to a new laboratory environment for one week before use. The animals were housed at 22 ± 1 °C and 55 ± 5% humidity on an auto-cycling 12 h light/dark cycle with free access to food and water in the Animal Center of Jeonbuk National University School of Dentistry during the experimental period. The use of mice in this study followed strict accordance with the recommendations in the Guide for the Animal Care and Use of the Jeonbuk National University. Experimental protocols were approved by the University Committee on Ethics in the Care and Use of Laboratory Animals.

### 2.6. Creation of Calvarial Defects 

A circular bone defect (4 mm in diameter) was created at the middle of the sagittal suture of mice (seven weeks old) according to methods described previously [11,12,17]. 

### 2.7. Implantation of ACSs or rhEGF-Loaded ACSs into the Defects

ACSs were cut into a circular form (4 mm diameter and 1 mm thickness) and impregnated with 15 μL of Dulbecco’s PBS only (ACS group) or supplemented with 5 µg of p-rhEGF (ACS + p-rhEGF group) or e-rhEGF (ACS + e-rhEGF group) for 10 min. Calvarial defects were implanted with each of ACSs and sutured with 6–0 chromic gut and 4–0 silk for internal and external surgical sites, respectively. At various times (0–10 weeks) of implantation with ACSs, the levels of newly formed bones in the region of calvarial defects were evaluated. 

### 2.8. Live μCT Analysis 

Live μCT scanning and image analyses (*n* = 6/group) were performed at 7 and 10 weeks post-surgery, in which all procedures for the μCT analysis followed the methods described elsewhere [12,17]. Briefly, the SkyScan NRecon Reconstruction package (Data viewer, Bruker-micro CT-Analyser Ver. 1.13 and CT Vol) was used to reconstruct 3D images, by which various bone-specific parameters, including bone volume (BV, mm^3^), bone volume percentage (BV/TV, %), bone surface (BS, mm^2^), BS in a total tissue volume (BS/TV, 1/mm), and bone mineral density (BMD, g/cm^3^), were calculated in newly formed bone in the defected regions. Here, BMD (g/cm^3^) was calculated by converting the attenuation data for volume of interest into Hounsfield units and BMD units using phantoms (SkyScan) that had a standard density corresponding to mouse bone. 

### 2.9. Immunohistochemistry (IHC) and Masson’s Trichrome (MT) Staining 

For IHC and Masson’s trichrome staining, tissue samples including the calvarial defect region were collected from the mouse groups (*n* = 6/group) at 2 and 4 weeks post-surgery and sectioned into 5 μm thicknesses, as described previously [11,12]. An IHC assay was performed using an ImmunoVectastain ABC kit (Vector Laboratories, Inc., Burlingame, CA, USA), where antibodies specific to Runx2, OCN, and OPN were used at 1:200 dilutions. Portions of the tissue sections were stained with MT as described previously [11,12]. Images of tissue samples were observed under a light microscope (Carl Zeiss, Oberkochen, Germany). 

### 2.10. Real-Time Reverse Transcription–Polymerase Chain Reaction (RT-PCR)

Mice (*n* = 6/group) were sacrificed at 2 weeks post-surgery, and the implanted ACSs were collected to isolate total RNA. The expression patterns of osteogenic marker genes including *Runx2*, *osterix*, *Ocn*, *Opn*, bone sialoprotein (*Bsp*), and type 1 collagen (*Col1a1*) were analyzed via real-time RT-PCR. The oligonucleotide primers specific to these genes were designed to amplify products less than 200 bp in length using Primer Express 3.0 (Applied Biosystems, Foster City, CA, USA), as shown in Appendix A. The thermocycling conditions for predenaturation and amplification with three-step cycles followed the methods described previously [12]. The *GAPDH* level was also monitored as the endogenous marker gene for quantification. 

### 2.11. In Vitro Evaluations on the p-rhEGF-Stimulated Cellular Responses

We evaluated in vitro effects of p-rhEGF on proliferation, the secretion of BMP2 and vascular endothelial growth factor (VEGF), and the expression of interleukin 8 (*IL-8*) in hPDL cells. The hPDL cells were collected from healthy patients (18–24 years old) who required tooth extraction prior to orthodontic treatment at Jeonbuk National University Dental Hospital (Jeonju, Republic of Korea). The donors provided written informed consent for use of their tissues, and the study was approved by the Ethical Committee of the Jeonbuk National University Hospital. The hPDL cells were grown in α-minimum essential medium (α-MEM) supplemented with 10% FBS and antibiotics (100 IU/mL penicillin G and 100 μg/mL streptomycin) in 60 mm culture dishes at 37 °C in a humidified atmosphere of 5% CO_2_. For the proliferation assay, the hPDL cells at the third passage were harvested and divided onto 96-well culture plates (5 × 10^3^ cells/well). After 12 h of incubation, cells were exposed to various concentrations (0, 5, 10, and 20 ng/mL) of p-rhEGF in growth medium. After an additional 24 h of incubation, the proliferation rate of the hPDL cells was determined using a Cell Counting Kit-8 (CCK-8) (Dojindo Molecular Tech., Kumamoto, Japan). Alternatively, hPDL cells were spread onto 24-well culture plates (5 × 10^5^ cells/well) in growth medium. The cells were exposed to p-rhEGF (0 to 20 ng/mL) for 24 h, and then, culture supernatants were harvested. The protein levels of BMP2 and VEGF were measured using human-anti-BMP2 (ab119581) and human-anti-VEGF ELISA Kits (ab222510). All experimental procedures for the CCK assay and ELISA followed the manufacturer’s instructions. Additionally, hPDL cells were incubated in the presence and absence of p-rhEGF (0 to 20 ng/mL) in 6-well culture plates. After 24 h of incubation, total RNA was extracted from the cells, and real time RT-PCR was performed to evaluate the expression of *IL-8* as described above. 

### 2.12. Statistical Analyses

All results are expressed as mean ± standard deviation (SD). One-way analysis of variance (ANOVA) was used to determine the significance of differences between more than two groups using Statistical Package for the Social Sciences (SPSS) (version 12.0). When one-way ANOVA was significant (*p* < 0.05), the post hoc Tukey test was used to determine significance differences among groups. Unpaired Student’s *t*-test was applied when the significance of differences between two sets of data was determined using GraphPad Prism V9 software. A *p* value < 0.05 was considered statistically significant.

## 3. Results

### 3.1. Characterization of ACSs

We observed the surface of ACSs by FE-SEM, in which no characteristic differences were found in morphological phenotypes such as filopodium and porosity between the ACSs alone and p-rhEGF-loaded ACSs (Figure 3A). Both the ACSs and p-rhEGF-loaded ACSs exhibited a correlation between compression stress (MPa) and strain (%) with a similar pattern (Figure 3B). The mechanical strength of the ACSs was also comparable with that of the p-rhEGF-loaded ACSs (Figure 3C). Similarly, the degradation rate of the ACSs in PBS did not differ from that of the p-rhEGF-loaded ACSs at a significant level (Figure 3D). These results indicate that the physiochemical properties of ACSs are not changed by impregnating rhEGF. Our findings also suggest that the structure of ACSs can be maintained for at least more than 4 weeks in defects after implantation.

### 3.2. Implanting the p-rhEGF-Impregnated ACSs Promotes New Bone Formation in Calvarial Defects Greater Than ACSs Alone Do

We evaluated whether the p-rhEGF is biologically active in stimulating bone formation using a mouse model of calvarial defects. Similar to our previous findings [11,12], the sham group which received surgical operation only did not show any new bone formation in the region of calvarial defects, indicating a critical-sized bone defect (data not shown). We next evaluated whether implantation with p-rhEGF-loaded ACSs enhances bone regeneration in the calvarial defects more than ACS implantation alone does through live μCT analysis at 7 and 10 weeks post-surgery. The 2D μCT images of the mouse groups indicated that the ACS + p-rhEGF group showed greater new bone formation in the defects in a time-dependent manner than the ACS group did (Figure 4A). The 2D μCT images also showed that the level of p-rhEGF-enhanced bone formation in the defects was similar to that of the e-rhEGF-stimulated bone healing. The reconstruction of 3D images (Figure 4B) and the bone parameter values at 7 (Figure 4C) and 10 weeks post-implantation (Figure 4D) supported the rhEGF-stimulated healing of the defective calvarial bones. 

As shown in the 3D images, implanting the p-rhEGF- or e-rhEGF-loaded ACSs increased further visible bone formation in the defects with similar activity compared with the ACS group. BV, BV/TV, BS, BS/TV, and BMD in the ACS + p-rhEGF or ACS + e-rhEGF group at 10 weeks post-implantation were significantly higher than those in the ACS group. To verify the effect of rhEGFs on defective bone healing, we isolated the ACSs implanted into calvarial defects after 4 weeks of implantation and stained them with MT. As shown in Figure 5A, the ACS + p-rhEGF and ACS + e-rhEGF groups exhibited greater regions positively stained with MT than the ACS group did. While the ACS group showed the collagenous (blue) and/or mineralized regions (red) mostly at the boundary of the defects, the rhEGF-delivered groups revealed such bone formation in whole regions of the defects. When the area (%) of blue-stained ACSs was calculated using an ImageJ program, the ACS group revealed significantly lower levels compared with those of the ACS + p-rhEGF or ACS + e-rhEGF group (Figure 5B).

### 3.3. Enhanced New Bone Formation in the ACS + p-rhEGF Group Is Correlated with the Increased Expression of Osteogenic Marker Molecules

We analyzed the protein levels of Runx2, OPN, and OCN in newly formed bone of calvarial defects 2 weeks after surgery via an IHC assay. Similar to the ACS + e-rhEGF group, the ACS + p-rhEGF group showed a greater induction of these osteogenic markers at the defect site compared with the ACS group (Figure 6A). The areas (%) of the protein-positive cells were also significantly greater in the ACS + p-rhEGF or ACS + e-rhEGF group compared with the ACS group (Figure 6B).

The results from real-time RT-PCR at 2 weeks post-implantation supported the rhEGF-stimulated upregulation of osteogenic marker genes, such that both the ACS + p-rhEGF and ACS + e-rhEGF groups showed significantly higher levels of *Runx2*, *osterix*, *Bsp*, *Col1a1*, *Ocn*, and *Opn* in the defects compared with the ACS control group (Figure 7).

### 3.4. Exogenous Addition of p-rhEGF Augments Proliferation and Induction of IL-8, BMP2, and VEGF in hPDL Cells

We further evaluated whether the p-rhEGF is a biologically active protein in stimulating EGF-signal-related cellular responses in hPDL cells. The exogenous addition of p-rhEGF significantly enhanced the proliferation rate of hPDL cells (Figure 8A). The expression of *IL-8*, an angiogenic chemokine, in the cells was also significantly increased by treating p-rhEGF (Figure 8B). When hPDL cells were exposed to p-rhEGF for 24 h, the protein levels of BMP2 (Figure 8C) and VEGF (Figure 8D) in culture supernatants were augmented in a dose-dependent manner. Specifically, the BMP2 level in untreated control cells (128 ± 8 pg/mL) was augmented up to 563 ± 35 pg/mL by treating them with 20 ng/mL p-rhEGF. Similarly, treatment with 20 ng/mL p-rhEGF induced an approximate 4.6-fold increase in VEGF in the cells, compared with the untreated control cells (182 ± 21 pg/mL).

## 4. Discussion

One of the main findings in this study is the successful production of rhEGF using a plant cell culture system. Various heterologous expression systems, including *E. coli*, mammalian, and yeast systems, have been developed to produce recombinant bioactive proteins [18,19], but these systems also exhibit several disadvantages derived from the protein folding, production costs, and structural differences in the glycosylation patterns of the products [20]. Transgenic plant cell suspension culture systems are considered to overcome disadvantages derived from those systems, as well as to preserve recombinant growth factors in a mature form with biologically active conformations [21]. Numerous studies have demonstrated that plant-cell-based expression systems utilize the protein storage organs and post-translational modification patterns similar to those of humans [22,23] and allow the large-scale production of recombinant proteins with a convenient isolation process [24,25]. Our current findings, along with previous studies [11,12], strongly support the concept that the plant cell suspension culture system using the RAmy3D promoter is a convenient and powerful system in producing recombinant growth factors. Considering the biological properties of EGF, our results also indicate the clinical usefulness of p-rhEGF in the healing of defective tissues such as skin and periodontal soft tissues.

In addition to the successful production of recombinant proteins using a plant-based system, the in vivo bioactivity of the products is also an important factor. The bioactivity of recombinant proteins is often evaluated by measuring their synergistic effects to enhance large or critical-sized bone defects in combination with a bioscaffold. Among various models of bone defects, calvarial defect models can be conveniently produced with various-sized defects, in which scaffolds impregnated with growth factors or in combination with other active molecules are easily delivered into the defected regions. Our results from the μCT analysis and MT staining support the concept that the local delivery of p-rhEGF-loaded ACSs induces the greater formation of new bones in defected regions than that of ACSs alone does. Our findings also show that the mass of newly formed bones in the ACS + p-rhEGF group is similar to the level in the ACS + e-rhEGF group. Moreover, the results from IHC and PCR assays reveal that the local implantation of p-rhEGF-loaded ACSs enhances bone formation in calvarial defects through the upregulation of osteogenic-specific markers such as Runx2, osterix, BSP, Col1a1, OCN, and OPN. 

In addition, it is important to understand whether the loading of growth factors affects the structural properties of scaffolds. Regarding this, our findings reveal that the surface of the ACS was not changed by impregnating rhEGF. Similar to our previous report [26], the impregnation of rhEGF did not alter physicochemical properties such as mechanical strength and degradation rate of ACSs. It is assumed that dissimilar to phenolic acids, recombinant proteins themselves do not increase mechanical strength of scaffolds, because of a week formation of carboxyamide bonds between the -COOH part of rhEGF and the -NH_2_ part of collagen [26]. This may also indicate that the rhEGF-mediated synergistic enhancement of defective bone healing is more closely associated with the nature of the factor to activate cellular response rather than changes in morphological and mechanical properties. 

Runx2 binds to the promoter regions of osteoblast-specific genes and controls the expression of downstream osteogenic markers, such as BSP, Col1a1, OCN, and OPN [27,28]. Runx2 is also essential for skeletal morphogenesis and cell cycle progression through interaction with proliferation-related genes [29,30]. Thus, the upregulation of Runx2 along with the attendant activation of its downstream molecules is to be the general mechanism in the process for defective bone healing. Our results support the concept that Runx2-mediated signaling is critical for the rhEGF-promoted healing of calvarial defects. Although there are opposite findings showing a negative correlation between EGF- and Runx2-mediated signaling [31,32], our findings suggest that the ACS-induced healing of defective bones might be facilitated via combination with p-rhEGF. 

To further verify the biological activity of p-rhEGF, we examined its effects on the proliferation and induction of BMP2, VEGF, and IL-8 in hPDL cells that express EGFR [33]. Our results show that a direct addition of p-rhEGF not only stimulates the proliferation of hPDL cells but also augments the secretion or expression of the osteogenic and angiogenic factors in the cells. There is a report showing that supplemental rhEGF enhances the production of VEGF and facilitates angiogenic response in bone marrow stem cells [34]. EGF also induces the production of IL-8, which protects endothelial cells against oxidative stress [35,36]. In addition, the increased expression of EGF can activate PDL cells for the healing of wounded PDL tissue by stimulating the induction of BMP2 and angiogenesis-associated molecules [37]. Furthermore, the local supplementation of rhEGF using a liposome or other vehicles facilitates the processes essential for bone regeneration and induces synergistic effects when given together with other growth factors [38,39]. Collectively, we consider that in addition to osteogenic stimulation through the activation of Runx2 and its downstream effectors, the p-rhEGF also triggers cellular proliferation, migration, and angiogenesis, from which a bone-defect healing can be synergistically enhanced. 

## 5. Conclusions

Our findings highlight that the transgenic rice calli suspension culture with the RAmy3D promoter is a powerful and efficient system in producing biologically active rhEGF protein. Our results support the concept that p-rhEGF exerts biopotencies in facilitating defective bone healing and in activating osteogenic and angiogenic responses that are critical events for wound healing. This study also demonstrates that the loading of rhEGF does not change the morphological and physicochemical characteristics of ACSs. Overall, this study indicates a clinical advantage of the p-rhEGF-loaded bioscaffolds in defective bone healing, as well as in activating osteogenic and angiogenic cellular responses.

## Figures and Tables

**Figure 1 materials-16-03335-f001:**
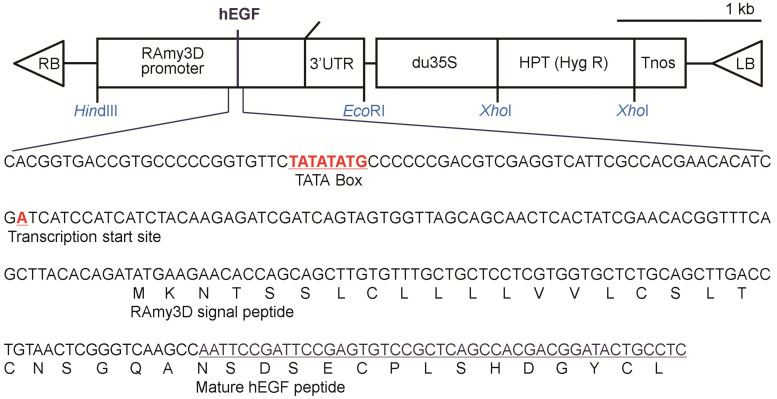
A schematic diagram of the gene construct applied to produce p-rhEGF. The human EGF gene, harboring the signal peptide of rice amylase 3D gene, is located between the rice amylase 3D promoter and the 3′untranslated region (3′ UTR). Transferred DNA (T-DNA) of the final plasmid is shown. RB, T-DNA right border; 3′ UTR, 3′ untranslated region of the rice α-amylase 3D gene; du35S, CaMV35S promoter with a duplicated enhancer region; HPT, hygromycin phosphotransferase (Hyg R); Tnos, terminator of nopaline synthase; LB, T-DNA left border.

**Figure 2 materials-16-03335-f002:**
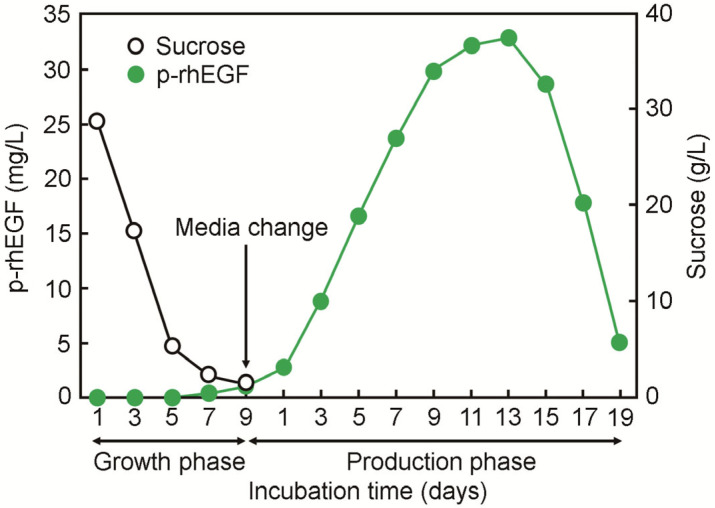
Time-dependent production of p-rhEGF in transgenic rice suspension culture medium. The selected rice cell line (S131-2 callus) was incubated in sucrose-containing N6 medium for 9 days, and then, the medium was replaced to sucrose-free N6 medium followed by additional incubation for 19 days. At the indicated days of incubation, p-rhEGF level was monitored via ELISA, in which day 13 was selected as the optimum incubation time.

**Figure 3 materials-16-03335-f003:**
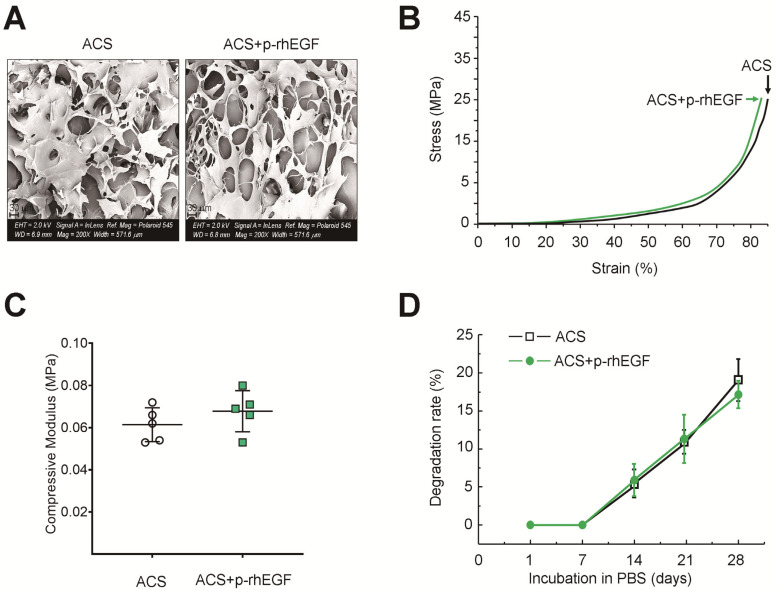
Physicochemical characterization of ACSs. (**A**) FE-SEM images showing the surface of ACSs. (**B**) Compressive stress (MPa)–strain (%) curves of ACSs and p-rhEGF-loaded ACSs under uniaxial loading. (**C**) The mean compressive moduli of ACSs (*n* = 5). (**D**) Degradation rate (%) of ACSs was determined after the indicated days of incubation in PBS (*n* = 4).

**Figure 4 materials-16-03335-f004:**
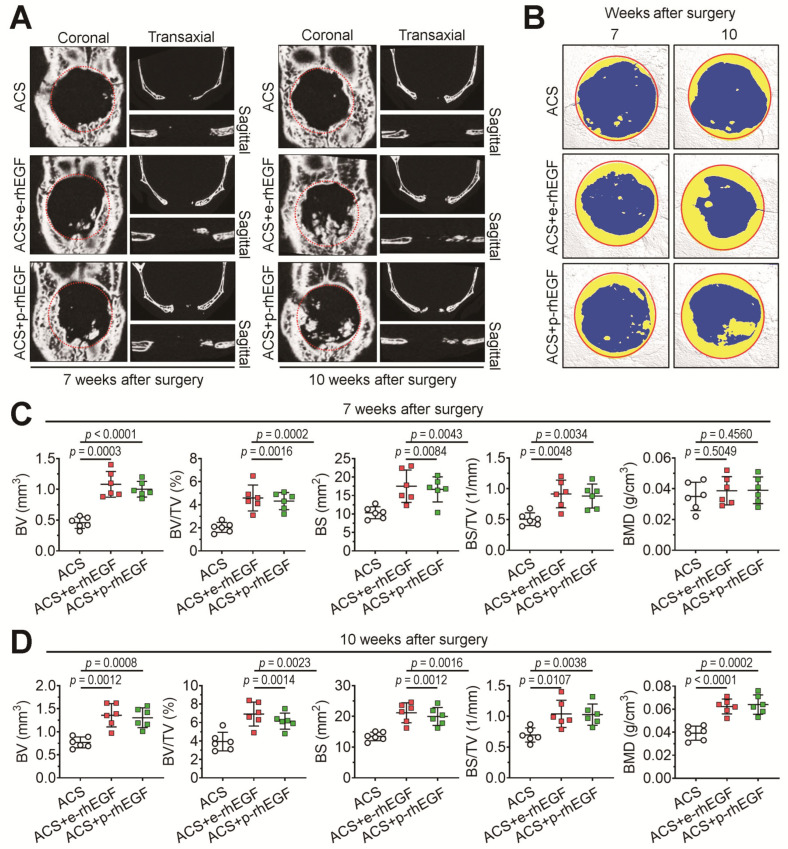
The loading of ACSs with p-rhEGF enhances defective bone healing in animal model of calvarial defects. (**A**) Experimental groups (ACS, ACS + e-rhEGF, and ACS + p-rhEGF groups) were subjected to live 2D μCT analysis 7 and 10 weeks after surgical operation. (**B**) Based on 2D μCT images, 3D images including calvarial defect regions were constructed. Red circles indicate the defected positions with the dimensions in 4 mm defect, while the yellow color designates newly formed bones within the margin of the defect. Values of bone parameters including bone volume (BV), bone volume percentage (BV/TV), bone surface (BS), BS in a total tissue volume (BS/TV), and bone mineral density (BMD) were measured in the regions of calvarial defects based on the 3D construction images (**C**) 7 and (**D**) 10 weeks after surgery (*n* = 6). Red and green squares in panels (**C**) and (**D**) indicate ACS+e-rhEGF and ACS+p-rhEGF groups, respectively. *p* values between the groups were determined using unpaired Student’s *t*-test.

**Figure 5 materials-16-03335-f005:**
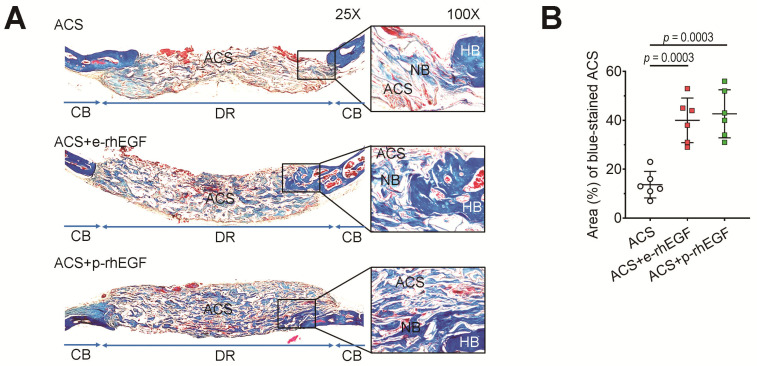
Implantation of p-rhEGF-loaded ACSs promotes the maturation of newly formed bones more greatly than that of ACSs alone does. (**A**) The maturity of newly formed bone in the defected regions was evaluated via Masson trichrome (MT) staining at 4 weeks post-surgery. The photographs show whole images of implanted scaffolds, in which the indicated rectangle regions were 4-fold magnified. In the defected regions, white color indicates non-mineralized ACS, whereas blue and red colors exhibit collagenous immature and matured bone formation in the ACS, respectively. NB, new bone; HB, host bone; CB, cortical bone; DR, defected region. (**B**) Area (%) of blue-stained ACSs after MT staining was evaluated using ImageJ software (Ver. 1.51, NIH, Bethesda, MD, USA; *n* = 6). *p* values between the groups were determined via unpaired Student’s *t*-test.

**Figure 6 materials-16-03335-f006:**
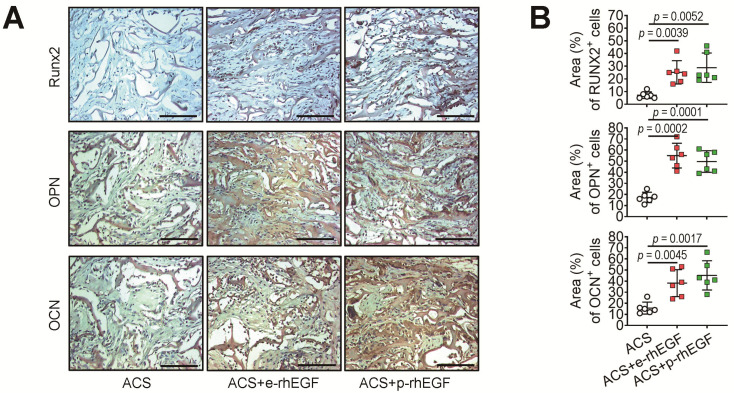
Impregnation of ACSs with rhEGFs augments the induction of osteoblast-specific marker proteins at the defected regions. (**A**) Two weeks after surgery, protein levels of Runx2, OPN, and OCN in the defected regions of the ACS, ACS + p-rhEGF, and ACS + e-rhEGF groups were determined via IHC staining. Bar, 100 μm; magnification, 200×. (**B**) Areas (%) of the cells stained with the marker-specific antibodies (shown as brown and/or dark-brown colors) were determined (*n* = 6). *p* values between the groups were determined via unpaired Student’s *t*-test.

**Figure 7 materials-16-03335-f007:**
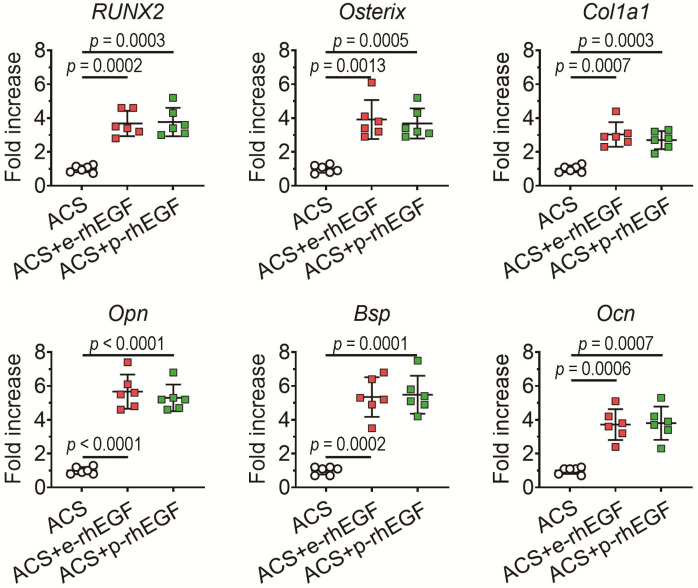
The rhEGF-stimulated increase in defective bone healing is accompanied by the upregulation of osteoblast-specific genes within the defected regions. Mouse groups were sacrificed 2 weeks after surgery, and expression levels of *Runx2*, *osterix*, *Bsp*, *Col1a1*, *Ocn*, and *Opn* in the implanted scaffolds were analyzed via real-time RT-PCR (*n* = 6). Red and green squares indicate ACS+e-rhEGF and ACS+p-rhEGF groups, respectively. *p* values between the groups were determined via unpaired Student’s *t*-test.

**Figure 8 materials-16-03335-f008:**
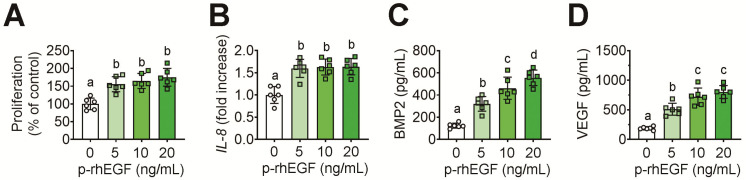
Exogenous addition of p-rhEGF stimulates proliferation, secretion of BMP2 and VEGF, and expression of *IL-8* in hPDL cells. The hPDL cells were exposed to the indicated concentrations of p-rhEGF for 24 h. (**A**) Proliferation rate, (**B**) mRNA expression of IL-8, and levels of (**C**) BMP2 and (**D**) VEGF secreted into culture supernatants were determined via CCK-8, real-time RT-PCR, and ELISA, respectively. The different superscripts ^(a–d)^ indicate significant differences (*p* < 0.05) among the groups by ANOVA (*n* = 5).

## Data Availability

The data used to support the findings of this study are available from the corresponding authors upon request.

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
