# Peer review of "Biopotentials of Collagen Scaffold Impregnated with Plant-Cell-Derived Epidermal Growth Factor in Defective Bone Healing"

_materials, 2023, doi:10.3390/ma16093335_

Round 1

Reviewer 1 Report

The manuscript "Biopotentials of Collagen Scaffold Impregnated with Plant Cell-Derived Epidermal Growth Factor in Defective Bone Healing" reports a new method for producing recombinant human epidermal growth factor (rhEGF) from a plant cell suspension culture system, and a histological evaluation of rhEGF-impregnated collagen scaffolds in a mouse model of calvarial defects. This manuscript demonstrates a useful strategy for growth factor-based bone regeneration, which is supported by the successful large scale production of plant-derived rhEGF with efficacy comparable to that of Escherichia coli-derived rhEGF. The manuscript is well-written and provides a clear insight into developing novel bone regenerative materials. Therefore, the reviewer believes that this manuscript can be acceptable for publication in Materials with some minor revisions.

Comment #1 

Image analysis on histological evaluation: The definition of "Area (%) of ACS stained with MT" is not very clear to the reviewer. In general protocols, Masson's trichrome staining produces blue collagen in bone and red mature (densely mineralized) bone as well as dark brown to black cell nuclei. Do the area percentages presented in the manuscript also include the portion of areas not assigned as bone (e.g., areas occupied by cells)? The authors are suggested specifically providing the definition of quantification for MT-stained images.

Comment #2

Release profiles and drug content: The osteogenic potential of the rhEGF-loaded collagen scaffolds should rely on release profile of rhEGF protein over time period of biodegradation and its loaded amount; these mainly depend on the binding affinity and architecture of scaffolds. Although controlling the release profile is not the central issue of this work, it could be encouraged to mention the putative mechanisms by which impregnated rhEGF proteins improved the bone regeneration over 4 to 10 week post-surgery. The authors are also suggested writing the immersion time of collagen scaffolds in rhEGF solution for impregnation.

Comment #3

Fabrication of collagen scaffolds: it is stated that the absorbable collagen scaffolds (ACSs) were prepared in accordance with the previously reported method [ref. 17]. This fabrication method described in the cited paper [17] yields coprecipitates of collagen and glycosaminoglycan (chondroitin sulfate). However, the involvement of chondroitin sulfate is not declared in this manuscript. Is the ACS fabricated in this study a purely collagen scaffold? The authors are suggested providing more details of the material fabrication.

Comment #4

Potential medical applications: Although the rhEGF-loaded ACSs are considered effective for bone regenerative therapy, the clinical advantages of this type of materials over other scaffolds (e.g., ceramic bone substitutes) are not clearly reviewed. Thus, the authors are recommended briefly mentioning potential medical applications of rhEGF-loaded ACSs based on its unique material properties.

Author Response

Reviewer #1

Comments and Suggestions for Authors

The manuscript "Biopotentials of Collagen Scaffold Impregnated with Plant Cell-Derived Epidermal Growth Factor in Defective Bone Healing" reports a new method for producing recombinant human epidermal growth factor (rhEGF) from a plant cell suspension culture system, and a histological evaluation of rhEGF-impregnated collagen scaffolds in a mouse model of calvarial defects. This manuscript demonstrates a useful strategy for growth factor-based bone regeneration, which is supported by the successful large scale production of plant-derived rhEGF with efficacy comparable to that of Escherichia coli-derived rhEGF. The manuscript is well-written and provides a clear insight into developing novel bone regenerative materials. Therefore, the reviewer believes that this manuscript can be acceptable for publication in Materials with some minor revisions.

Comment #1 

Image analysis on histological evaluation: The definition of "Area (%) of ACS stained with MT" is not very clear to the reviewer. In general protocols, Masson's trichrome staining produces blue collagen in bone and red mature (densely mineralized) bone as well as dark brown to black cell nuclei. Do the area percentages presented in the manuscript also include the portion of areas not assigned as bone (e.g., areas occupied by cells)? The authors are suggested specifically providing the definition of quantification for MT-stained images.

â–ºAuthor response: Thank you for the reviewers’ valuable comments on the definition of MT staining. Actually, the figure 4B (the Area (%) of ACS stained with MT/Figure 5B in the revised manuscript) was the area of blue region shown in the MT-stained ACS. The original indication of Y-axis was wrong. In the revised version, we changed the Y-axis as “Area (%) of blue-stained ACS”, as well as revised the related sentences. Based on the properties of MT staining, we also revised the Results section to provide further exactly the meaning of MT staining-derived results.   

Comment #2

Release profiles and drug content: The osteogenic potential of the rhEGF-loaded collagen scaffolds should rely on release profile of rhEGF protein over time period of biodegradation and its loaded amount; these mainly depend on the binding affinity and architecture of scaffolds. Although controlling the release profile is not the central issue of this work, it could be encouraged to mention the putative mechanisms by which impregnated rhEGF proteins improved the bone regeneration over 4 to 10 week post-surgery. The authors are also suggested writing the immersion time of collagen scaffolds in rhEGF solution for impregnation.

 â–ºAuthor response: The authors deeply agreed with the reviewers’ comments on the putative mechanisms derived from the binding activity and architecture of scaffolds. Although we can’t exactly state such mechanisms, considering our previous study (Bhattarai et al., Biomaterials Advances, 2022;135:112673; Ref 26 in the revised manuscript) suggests that there might be the formation of carboxyamide bond between -COOH part of EGF and -NH2 part of collagen. We added this statement with the related reference in the revised version. In relation to another reviewers’ suggestion, we also provided several physiochemical properties of ACSs in the revised manuscript (as Figure 3). We consider that the additional statements with new data further help the readers to be understood. In addition, we provided the time (10 min) to impregnate rhEGF into ACSs in the section of Materials and methods.

Comment #3

Fabrication of collagen scaffolds: it is stated that the absorbable collagen scaffolds (ACSs) were prepared in accordance with the previously reported method [ref. 17]. This fabrication method described in the cited paper [17] yields coprecipitates of collagen and glycosaminoglycan (chondroitin sulfate). However, the involvement of chondroitin sulfate is not declared in this manuscript. Is the ACS fabricated in this study a purely collagen scaffold? The authors are suggested providing more details of the material fabrication.

â–ºAuthor response: The authors understood on the reviewers’ comments. We consider that the original information for the ACS fabrication may evoke some confused understanding as like the reviewer commented. To clarify the methodology to fabricate ACS, we provided further detail information on the material fabrication, through which we can also ensure that the ACSs used in this study will be free of a side chemical contamination.

Comment #4

Potential medical applications: Although the rhEGF-loaded ACSs are considered effective for bone regenerative therapy, the clinical advantages of this type of materials over other scaffolds (e.g., ceramic bone substitutes) are not clearly reviewed. Thus, the authors are recommended briefly mentioning potential medical applications of rhEGF-loaded ACSs based on its unique material properties.

â–ºAuthor response: Thank you for the helpful comments. Regarding to the reviewers’ comments, we added several statements in the Introduction and Discussion sections, from which the advantage of p-rhEGF might be further considered.

All changes in the revised manuscript were highlighted in red. We also added several data (Figure 3 in the revised version) to characterize the physiochemical properties of ACSs. We hope that this revision satisfies the requirements of the reviewer.

Reviewer 2 Report

The work was conducted by  Poudel et al dedicating to production of recombination human epidermal growth factor by using plant cells. The synthesized protein then was incorporated into a collagen scaffold and successfully applied for calvarial defect regeneration. In summary, the study is quite useful and interesting. However, the content of manuscript must be complemented intensely before it is considered for publication.

1. The introduction and conclusion parts are too concise to well understand the research, I suggesting the authors should be improved, for example, the manuscript can add content about tissue engineering, role of scaffold and growth factor in tissue engineering, previous studies using plant cells derived growth factors for bone regeneration.

2. There is large amount of researches in production of rhEGF using plant cells, so authors need to mention advantages of rice calli cell culture system in compare with other plant cells in term of yield, feasibility.

3. For preparation of ACSs, even though authors followed from the another, the procedure need to describe in brief and show the chemical structure of materials.

4. I suggest the authors adding characterization of ACSs, at least with morphology (SEM analysis), FTIR spectrum, porosity and compression strength.

Author Response

Reviewer #2

Comments and Suggestions for Authors

The work was conducted by Poudel et al dedicating to production of recombination human epidermal growth factor by using plant cells. The synthesized protein then was incorporated into a collagen scaffold and successfully applied for calvarial defect regeneration. In summary, the study is quite useful and interesting. However, the content of manuscript must be complemented intensely before it is considered for publication.

  1. The introduction and conclusion parts are too concise to well understand the research, I suggesting the authors should be improved, for example, the manuscript can add content about tissue engineering, role of scaffold and growth factor in tissue engineering, previous studies using plant cells derived growth factors for bone regeneration.

â–ºAuthor response: Thank you for the helpful comments on the Introduction section. In the revised manuscript, we provided further information on the combination of biomaterials including growth factors in bone tissue engineering. The application of plant cell-derived growth factors (such as rhIGF and rhFGF) for bone regeneration are also included in the section.

  1. There is large amount of researches in production of rhEGF using plant cells, so authors need to mention advantages of rice calli cell culture system in compare with other plant cells in term of yield, feasibility.

â–ºAuthor response: Based on the reviewers’ comments, we have added the advantage of rice callus cell culture system comparing that with other plant culture or system in the Introduction section.

  1. For preparation of ACSs, even though authors followed from the another, the procedure need to describe in brief and show the chemical structure of materials.

â–ºAuthor response: The authors deeply understood on the reviewers’ comment. In the revised version, we provided further detail information in fabricating ACSs. Although we didn’t show the chemical structure of materials, a possible linkage between rhEGF and ACS was suggested in the revised version, in which we consider that there might be the formation of carboxyamide bond between -COOH part of EGF and -NH2 part of collagen based on our previous report [Ref 26 in the revised manuscript; Bhattarai et al., Biomaterials Advances, 2022;135:112673]. To further provide the characteristic properties of ACSs such as the surface and mechanical strength, we added several new data in Figure 3, from which we believe the revised version might improve the understanding on the fabrication and physiochemical properties of ACSs.

  1. I suggest the authors adding characterization of ACSs, at least with morphology (SEM analysis), FTIR spectrum, porosity and compression strength.

â–ºAuthor response: As the authors’ response shown above, we provided the data from FE-SEM, compression strength, and degradation of ACSs. We also added our previous report [Ref 26 in the revised version], where we had characterized such physiochemical properties of ACSs.

All changes in the revised manuscript were highlighted in red. We hope that this revision satisfies the requirements of the reviewer.

Reviewer 3 Report

The authors investigate the biopotency of Collagen Scaffold Impregnated with Plant Cell-Derived Epidermal Growth Factor for their use in Defective Bone Healing. The paper is interesting and the results reported are attractive for the possible scaffold applications, however there are several open questions and a number of changes which are needed before the paper can be ready for publication. Following, I have reported some suggestions for the authors. The paper must also be revised for English fluency and for many typing errors.

 Materials and Methods:

-More information is needed, that should be reported in a clear way in the manuscript. I think that it is important to report in this article the significant information to describe the absorbable collagen scaffold (ACS) fabrication method. Please rewrite paragraph and include this method. I also suggest to create the paragraph on “p-rhEGF-loaded absorbable collagen scaffold”, which is missing in the Materials and Methods section, to report a detailed protocol.

 -“Minimal data” are given in text and figure 3,4,5,6,7 legends. To make these attractive panel useful to the reader, please specify in the legends all acronym reported also adding the statistical analyses, with asterisks.

-The figure 3C and 3D are confusing and should be changed adding the time of evaluation, 7 and 10 weeks.

-Please make uniform the font and the size in whole manuscript.

Author Response

Reviewer #3

Comments and Suggestions for Authors

The authors investigate the biopotency of Collagen Scaffold Impregnated with Plant Cell-Derived Epidermal Growth Factor for their use in Defective Bone Healing. The paper is interesting and the results reported are attractive for the possible scaffold applications, however there are several open questions and a number of changes which are needed before the paper can be ready for publication. Following, I have reported some suggestions for the authors. The paper must also be revised for English fluency and for many typing errors.

 Materials and Methods:

-More information is needed, that should be reported in a clear way in the manuscript. I think that it is important to report in this article the significant information to describe the absorbable collagen scaffold (ACS) fabrication method. Please rewrite paragraph and include this method. I also suggest to create the paragraph on “p-rhEGF-loaded absorbable collagen scaffold”, which is missing in the Materials and Methods section, to report a detailed protocol.

â–ºAuthor response: Thank you for the helpful comments on the Materials and methods section. In the revised manuscript, we provided further detail methodology in the fabrication of ACSs. We also created a new method section for the implantation of ACS or ACSs loaded with rhEGF.

 -“Minimal data” are given in text and figure 3,4,5,6,7 legends. To make these attractive panel useful to the reader, please specify in the legends all acronym reported also adding the statistical analyses, with asterisks.

â–ºAuthor response: Based on the reviewers’ comments, we provided the explanation of acronym used in legends of the indicated figures if needed. In the case of statistical analyses, we showed the statistic values instead of the indication with asterisks. The authors consider that this provided further detail information on the results from statistical analyses. For example, p < 0.003 and p < 0.0003 indicate “**” and “***”, respectively, when the results are expressed with asterisks. In the revised manuscript, we provided the statistical methods such as “unpaired Student t-test” and “ANOVA” to further clarify how the statistic values are derived. 

-The figure 3C and 3D are confusing and should be changed adding the time of evaluation, 7 and 10 weeks.

â–ºAuthor response: To further clarify the meaning of figure 3C and D (Figure 4C and D in the revised manuscript), we revised the figures providing the weeks of evaluation on the top of each of figures.

-Please make uniform the font and the size in whole manuscript.

â–ºAuthor response: The authors understood on the reviewers’ comment on the font sizes. The authors consider that this problem was derived from the editorial process to change the originally submitted form to that used in the Journal. During this revision, we carefully checked all contents and unified the font sizes. We also have tried to correct all typo-errors in the revised manuscript.

In addition, all changes in the revised manuscript were highlighted in red. We also added several data as Figure 3 in the revised version to characterize the physiochemical properties of ACSs. We hope that this revision improves the quality of this paper and satisfies the requirements of the reviewer.

Reviewer 4 Report

The current study focuses on enhancing bone healing by combining scaffolds with recombinant human epidermal growth factor (rhEGF) protein through a synergistic activation to stimulate cell growth, differentiation, and survival. The rhEGF protein (p-rhEGF) was produced in plant cells by inserting a cloned expression vector containing its cDNA under the control of the rice -amylase 3D promoter into rice calli (Oryza sativa L. cv.Dongjin). In the criterion of callus-sized calvarial defects, implanting ACS loaded with p-rhEGF (5 g) improved new bone formation. Indeed, p-rhEGF production, as well as its ability to improve SCA-mediated bone regeneration by activating cellular responses required for wound healing. The experimental developments of the study have been carried out in a rigorous manner. The style is clear, and the content is rich and dense. It is based on a rigorous formalism of work. In conclusion, I express a very favorable opinion on the publication of this work.

Author Response

Reviewer #4

Comments and Suggestions for Authors

The current study focuses on enhancing bone healing by combining scaffolds with recombinant human epidermal growth factor (rhEGF) protein through a synergistic activation to stimulate cell growth, differentiation, and survival. The rhEGF protein (p-rhEGF) was produced in plant cells by inserting a cloned expression vector containing its cDNA under the control of the rice -amylase 3D promoter into rice calli (Oryza sativa L. cv.Dongjin). In the criterion of callus-sized calvarial defects, implanting ACS loaded with p-rhEGF (5 g) improved new bone formation. Indeed, p-rhEGF production, as well as its ability to improve SCA-mediated bone regeneration by activating cellular responses required for wound healing. The experimental developments of the study have been carried out in a rigorous manner. The style is clear, and the content is rich and dense. It is based on a rigorous formalism of work. In conclusion, I express a very favorable opinion on the publication of this work.

â–ºAuthor response: Thank you for the reviewers’ positive evaluation on this manuscript. Based on the comments of other reviewers, the authors have finely revised the manuscript by providing further detailed information in the sections of Introduction, Materials and methods, and Discussion, as well as by including several new data to characterize the physiochemical properties of ACSs. All changes in the revised manuscript were highlighted in red. We believe that this revision extensively improved the quality of this manuscript.

Round 2

Reviewer 2 Report

The authors have addressed all of my concerns, I recommend it for publication after English checking.